# Effect of Prenatal Yoga versus Moderate-Intensity Walking on Cardiorespiratory Adaptation to Acute Psychological Stress: Insights from Non-Invasive Beat-to-Beat Monitoring

**DOI:** 10.3390/s24051596

**Published:** 2024-02-29

**Authors:** Miha Lučovnik, Helmut K. Lackner, Ivan Žebeljan, Manfred G. Moertl, Izidora Vesenjak Dinevski, Adrian Mahlmann, Dejan Dinevski

**Affiliations:** 1Department of Perinatology, Division of Obstetrics and Gynaecology, University Medical Centre Ljubljana, 1000 Ljubljana, Slovenia; 2Faculty of Medicine, University of Ljubljana, 1000 Ljubljana, Slovenia; 3Otto Loewi Research Center, Division of Physiology and Pathophysiology, Medical University of Graz, 8036 Graz, Austria; 4Department for Women’s Health, Health Center Lenart, 2230 Lenart v Slovenskih Goricah, Slovenia; ivan.zebeljan@zd-lenart.si; 5Department of Philosophy, Alpen-Adria University Klagenfurt, 9020 Klagenfurt, Austria; manfredmoertl@gmx.at; 6Sončna Vila Yoga Studio, 2000 Maribor, Slovenia; izi@izidora.si; 7Department of Internal Medicine III, University Hospital Carl Gustav Carus at Technische Universität Dresden, 01307 Dresden, Germany; mahlmanna@kkh-hagen.de; 8Centre for Vascular Medicine, Clinic of Angiology, St.-Josefs-Hospital, Katholische Krankenhaus Hagen gem. GmbH, 58097 Hagen, Germany; 9Faculty of Medicine, University of Maribor, 2000 Maribor, Slovenia; dejan.dinevski@um.si

**Keywords:** pregnancy, yoga, heart rate variability, cardio-respiratory adaptation, autonomic nervous system, mental challenge

## Abstract

We recently reported enhanced parasympathetic activation at rest throughout pregnancy associated with regular yoga practice. The present study presents a secondary analysis of data collected within a prospective cohort study of 33 pregnant women practicing yoga once weekly throughout pregnancy and 36 controls not involved in formal pregnancy exercise programs. The objective was to assess the impact of prenatal yoga on the autonomic nervous system stress response. Healthy pregnant women with singleton pregnancies were recruited in the first trimester. There was no significant difference in the maternal body mass index (BMI) between the yoga group and the controls (24.06 ± 3.55 vs. 23.74 ± 3.43 kg/m^2^, *p* = 0.693). Women practicing yoga were older (28.6 ± 3.9 vs. 31.3 ± 3.5 years, *p* = 0.005) and more often nulliparous (26 (79%) vs. 18 (50%), *p* = 0.001). We studied heart rate variability (HRV) parameters in the time domain (SDNN, standard deviation of regular R-R intervals, and RMSSD, square root of mean squared differences of successive R-R intervals) and frequency domain (ln(LF/HF), natural logarithm of low-frequency to high-frequency power), as well as synchronization indices of heart rate, blood pressure and respiration during and immediately following acute psychological stress of a standardized mental challenge test. Measurements were performed once per trimester before and after yoga or a 30 min moderate-intensity walk. Statistical comparison was performed using three-way analyses of variance (*p* < 0.05 significant). Time domain HRV parameters during and following mental challenge in the yoga group were significantly higher compared to the controls regardless of the trimester (*F* = 7.22, *p* = 0.009 for SDNN and *F* = 9.57, *p* = 0.003 for RMSSD, respectively). We observed no significant differences in the yoga group vs. the controls in terms of ln(LF/HF) and synchronization indices. Regular prenatal yoga practice was associated with a significantly reduced sympathetic response to mental challenge and quicker recovery after acute psychological stress. These effects persisted throughout pregnancy with regular practice.

## 1. Introduction

Yoga is a body–mind practice that encompasses a system of body postures (asanas), combined with breathing (pranayama) and concentration (dharana) exercises as well as meditation (dhyana) techniques [1]. It has gained popularity in Western countries over the last century [1]. Research has identified yoga as one of the most commonly used methods for a complementary health approach (besides meditation and seeing a chiropractor) [2]. The percentage of adults in the United States who used yoga for this purpose in the last 12 months increased from 9.5% in 2012 to 14.3% in 2017 [3]. About 70% of practitioners are women, with the majority in their reproductive age [4]. As a result, yoga is increasingly popular among pregnant women [4,5]. In the US, 7% of women reported practicing yoga during pregnancy [4].

Yoga is often listed among the safest and most beneficial forms of physical activity during pregnancy [6]. Studies showed associations between prenatal yoga and decreased incidence of fetal growth restriction, preterm delivery, prolonged and dysfunctional labor as well as perinatal mental disorders such as antenatal anxiety, depression and stress [7,8,9,10,11,12,13]. This could be, at least in part, attributed to the beneficial effects of yoga on autonomous nerve activity. The maternal cardiovascular system undergoes profound changes during pregnancy and the autonomic nervous system plays a central role in cardiovascular adaptation to pregnancy-related hemodynamic changes [14,15,16,17,18]. Our group has recently reported enhanced parasympathetic activation at rest throughout pregnancy associated with regular yoga practice [19]. 

Yoga has also been reported to improve stress reactivity in healthy non-pregnant adults [20]. This could further enhance yoga’s beneficial effects on the maternal cardiovascular system. The impact of yoga on acute stress response in pregnancy has, however, not been studied yet. This manuscript presents a secondary analysis of data collected within a prospective cohort study of pregnant women practicing yoga regularly from the first trimester [19]. Data on alterations of heart rate variability (HRV) and cardio-respiratory phase synchronization during and immediately following acute stress response to mental challenge are original and have not been published previously. The objective of the present study was to explore the effect of yoga compared to moderate walking on HRV parameters and cardio-respiratory synchronization indices during and following acute stress using non-invasive beat-to-beat monitoring for hemodynamic and autonomic functions of the human body.

## 2. Materials and Methods

We included 69 healthy pregnant women in a prospective cohort study from August 2020 to April 2022. Women with preexisting cardiovascular disease (including hypertension and arrhythmias), taking medications that would affect heart rate or blood pressure, psychiatric disorders, epilepsy, kidney disease, liver disease, rheumatoid autoimmune disorders, diabetes mellitus, alcohol and/or illicit drug abuse, known fetal anomaly, or multiple pregnancies were excluded from the study. Previous yoga experience was not considered an exclusion criterion. Thirty-three women practiced pregnancy-adapted yoga according to the system Yoga in daily life regularly (at least once weekly) throughout all three trimesters of pregnancy [21]. Yoga classes were held at Soncna vila yoga studio in Maribor, Slovenia and were led by a certified experienced prenatal yoga instructor. Classes lasted for 90 min and consisted of initial relaxation (10–15 min), followed by yoga postures and stretching exercises and a final 20–30 min of breathing and meditation techniques. The adaptation of yoga practices for specific gestational age was based on in-depth consultations with gynecologists and experienced physical therapists. Thirty-six women (controls) did not attend any formal physical exercise program. All participants provided written informed consent to study participation. The study was approved by the Institutional Review Board of the University Medical Center Maribor [22] and the Slovenian National Medical Ethics Committee (project number 0120-575/2018/5, approved on 22 February 2019). The detailed participant enrollment procedure and study flow have been published previously [19] (Clinicaltrials.gov registration: NCT04476368).

The study protocol is presented in Figure 1. All measurements were performed in the afternoon, before and after yoga practice (or a 30 min moderate-intensity walk in the control group) once per trimester of pregnancy. Measurements in the yoga group were performed before and after yoga practice. Measurements in the control group were performed before and after 30 min of moderate-intensity walking. 

Mental challenge comprised of a standardized memory task. First, the participants had to memorize 12 character strings of 4 letters. This “learning” memory task was followed by 2 min of paced breathing at 10 min^−1^ and then by the memory task itself, during which the participants had to choose the right (memorized) character string out of 5 options. This mental challenge was followed by a rest/recovery phase (Figure 1).

Continuous monitoring of blood pressure (sampling rate (sr) = 100 Hz, blood pressure range = 50–250 mmHg, ±5 mmHg) and heart rate (R-R intervals derived from 3-lead electrocardiography (ECG), sr = 1 kHz, f_cut-off_ = 0.08–150 Hz) was carried out throughout the whole measurement protocol with the Task Force^®^ Monitor (CNSystems, Medizintechnik AG, Graz, Austria) [23]. Continuous blood pressure was measured using the participants’ finger and a refined version of the vascular unloading technique, which was corrected to absolute values with oscillometric blood pressure measurement by the Task Force^®^ Monitor [23]. Thoracic impedance electrodes were placed at the neck and thoracic regions (specifically in the midclavicular line at the xiphoid process level). Respiration was derived from thoracic impedance (sr = 500 Hz for thoracic impedance; for synchronization analysis, we used a resampled signal of 4 Hz). Raw data were exported to MATLAB^®^ (The Math Works, Inc., Natick, MA, USA) data format for further analysis. A semi-automatic artifact-handling software, described in detail in our previous publications, was used for artifact handling of the continuous recordings [19,24].

### 2.1. Heart Rate Variabity Parameters

The time domain HRV parameters analyzed during and following mental challenge (memory task and post-intervention recovery in Figure 1) were SDNN (standard deviation of the regular R-R intervals) and RMSSD (square root of the mean squared differences of successive R-R intervals). While the RMSSD is influenced more by vagal tone, the SDNN represents the activity of both sympathetic and parasympathetic branches.

The frequency domain HRV parameter analyzed was the ln(LF/HF) (natural logarithm of low-frequency (LF) to high-frequency (HF) power ratio) [25]. The prerequisites for the calculation of the HRV variables, such as the “quasi-stationarity” of the intervals and the appropriate estimation of the variables in the frequency range, were checked in advance using basic mathematical equations (e.g., Parseval’s theorem) [24].

### 2.2. Phase Synchronization Indices

The concept of analytic signals based on the Hilbert transform can be used to define the phase of an arbitrary signal such as the heart rate. This approach was implemented to calculate the phase synchronization indices, e.g., heart rate and systolic blood. The algorithm of our software toolbox, implemented in Matlab^®^ (R2022a), operates in the following way:
The function HILBERT compute the so-called discrete-time analytic signal X with X = X_r_ + I × X_i_ in a narrow frequency band such that X_i_ is the Hilbert transform of the real vector X_r_. To obtain a clear physical interpretation that is given only for narrow band signals, we use the band-pass filtered time series.In the next step, the function ANGLE is used to calculate the phase of the resulting signal X at every time point with the function.Subsequently, the difference between two given phase vectors for the interpolated bivariate data series, e.g., between heart rate and systolic blood pressure, can be calculated.The distribution of phase difference Ψ(t_i_) is quantified by the synchronization index γ defined as
γ = {cos Ψ(t_i_)}^2^ + {sin Ψ(t_i_)}^2^ [0 ≤ γ ≤ 1],
where the brackets {…} denote an average and t_i_ the sample times.

If the synchronization index γ = 1, then both time series are completely synchronized in a statistical sense, while in the case of γ = 0, both time series are completely desynchronized, i.e., the values of Ψ(t_i_) are equally distributed in the range of [−π, π]. Thus, phase synchronization provides a quantitative indicator of the coordinated behavior of pairs of systems. The methodology to calculate cardio-respiratory indices and the rationale behind these calculations have been described previously by our research group [17].

Continuous blood pressure was obtained using the participants’ finger as mentioned above. Therefore, an additional aspect must be taken into account: the continuous blood pressure signal needs to be “calibrated” via the blood pressure cuff on the upper arm at regular intervals. Nevertheless, for offline recalculations like our software, recalculation criteria can be defined to correctly estimate the dynamic behavior between two “calibration points”. We use pulse transition time as additional plausibility criteria in our updated software to ensure an optimal estimation of the time series of systolic (and diastolic) blood pressure.

Synchronization indices between systolic blood pressure and RR interval (inter-beat interval between successive heartbeats) (γ_SBP* × RR_), between respiratory frequency and RR interval (γ_RF × RR_), as well as between respiratory frequency and systolic blood pressure (γ_RF × SBP)_ were analyzed for the purpose of the study. 

### 2.3. Statistical Analysis

Statistical comparison of background clinical characteristics in yoga vs. control groups was performed using the univariate analysis of variance (F-test) for continuous variables and Chi-square test for categorical variables. For continuous variables, data were expressed as means with standard deviations. Categorical data were summarized as frequencies and percentages. To determine whether the intervention (*yoga* vs. *control*) significantly influenced a given parameter measured, the differences in HRV parameters and phase synchronization indices were statistically compared using 2 × 3 × 2 three-way analyses of variance. The delta values (e.g., SDNN after intervention [yoga or walk] minus heart rate before intervention) of time intervals (*memory task and recovery*) and trimester of pregnancy (*first*, *second and third trimester*) were the within-subject factors in these analyses, and intervention (*yoga* vs. *control*) was the between-subject factor, thus accounting for the physiological change in this parameter throughout pregnancy and following any physical activity. A *p* ≤ 0.05 was considered statistically significant and the effect size of yoga intervention was estimated using the partial Eta-squared (η_p_^2^). IBM SPSS Statistics for Windows Version 25.0 (IBM Corp., Armonk, NY, USA) was used for statistical analysis.

## 3. Results

All 69 women included in the study were of Caucasian ethnicity. There was no significant difference in the maternal body mass index (BMI) between the yoga group and controls (24.06 ± 3.55 vs. 23.74 ± 3.43 kg/m^2^, *p* = 0.693). Women practicing yoga were older (28.6 ± 3.9 vs. 31.3 ± 3.5 years, *p* = 0.005) and more often nulliparous (26 (79%) vs. 18 (50%), *p* = 0.001) [19]. 

Table 1 presents the changes in HRV parameters after minus before yoga practice (yoga group) and after minus before 30 min walk (control group) in all three trimesters of pregnancy. 

Between-subject analysis showed a significantly lower heart rate during and immediately following a mental challenge in the yoga group compared to controls in all three trimesters (*F* = 31.26, *p* < 0.001, η_p_^2^ = 0.32). We also observed significantly higher time domain HRV parameters during and following mental challenge in the yoga group vs. controls regardless of pregnancy trimester (*F* = 7.22, *p* = 0.009, η_p_^2^ = 0.10 for SDNN and *F* = 9.57, *p* = 0.003, η_p_^2^ = 0.13 for RMSSD, respectively). In terms of frequency domain HRV parameters, ln(LF/HF) did not significantly decrease during or following mental challenge in the yoga vs. control groups (*F* = 0.84, *p* = 0.363).

Within-subject analysis showed a significant interaction for time intervals (memory task and recovery phase) × group and three-time interaction for trimester × time intervals × group for RMSSD (*F* = 4.12, *p* = 0.046, η_p_^2^ = 0.06 and *F* = 4.27, *p* = 0.016, η_p_^2^ = 0.06, respectively), whereas no significant interactions were observed for HR, SDNN, or ln(LF/HF) (all *p*’s > 0.12 and >0.054, respectively) (Figure 2).

Table 2 presents blood pressure variables, respiratory frequency and phase synchronization indices after minus before yoga practice (yoga group) and after minus before 30 min walk (control group) in all three trimesters of pregnancy. 

Between-subject analysis showed no difference in systolic blood pressure (SBP) (*F* = 0.30, *p* = 0.587), mean arterial pressure (MAP) (*F* = 0.80, *p* = 0.376), diastolic blood pressure (DPB) (*F* = 0.81, *p* = 0.371), respiratory frequency (RF) (*F* = 2.21, *p* = 0.142), γ_RF × RR_ (*F* = 0.01, *p* = 0.944), or γ_RF × SBP_ (*F* = 0.27, *p* = 0.606) during or immediately following a mental challenge in the yoga group compared to controls in all three trimesters. There was a statistically significant difference towards a lower change in the synchronization index between the SBP and RR interval γ_SBP × RR_ in the yoga group compared to the controls (*F* = 4.59, *p* = 0.037, η_p_^2^ = 0.06).

Within-subject analysis showed no significant interaction for time intervals × group or three-time interaction for trimester × time intervals × group (all *p* values > 0.089 and > 0.061, respectively). 

## 4. Discussion

Women in the yoga group had significantly increased time domain HRV parameters (i.e., SDNN and RMSSD) during the memory task and during the recovery phase compared to the controls. 

The study adds important novel information to our previous report on significantly increased SDNN and RMSSD as well as decreased ln(LF/HF) at rest in women practicing prenatal yoga compared to pregnant controls not enrolled in formal pregnancy exercise programs [19]. The present analysis of non-invasive data acquired during the mental challenge and recovery phases of our measurement protocol shows that regular yoga practice during pregnancy results in improved stress reactivity with blunted sympathetic surges associated with acute stressful events as well as faster recovery from stress. These results are in accordance with several studies on the effects of yoga on the autonomic nervous system published to date by other groups. Practicing yoga has been shown to increase parasympathetic activity outside pregnancy in both novices and experienced practitioners [26,27,28,29,30,31]. Single yoga sessions have also been shown to improve stress reactivity in healthy non-pregnant adults [20]. In pregnancy, acute decreases in sympathetic activity not provoked by stress tests following single yoga classes have been reported by Satyapriya et al. [14]. As demonstrated in previous research, prenatal yoga practice was mostly associated with HRV changes while there was no significant impact of yoga on cardio-respiratory synchronization indices [14,19].

The neurovascular responses to psychological stress of a memory task are well known and include activation of the sympathetic system, increases in heart rate, cardiac output, blood pressure, vasoconstriction in the splanchnic and renal regions, and vasodilatation in skeletal muscles [18]. It has to be noted that response to memory tasks can be influenced by several factors, like the response option or the stimulus interval [32]. Nevertheless, in a recent review, Immanuel et al. concluded that HRV is a reliable measure of psychological stress response when standard stress induction and assessment protocols combined with validated HRV parameters in different domains are used [33]. They also stated that RMSSD was the most frequently reported HRV metric [33]. Several advantages of time domain HRV parameters have also recently been reported by our group [24]. In the present study, responses to memory-task-associated stress were assessed using a short and simple protocol with almost no influence of the response (a mouse click only) and stimulus interval (mouse clicks at irregular time intervals) on HRV parameters.

No study to date has focused on the effects of yoga on a pregnant woman’s ability to cope with acute psychological stress. Lackner et al. described a decrease in parasympathetic activity during mental challenge in healthy non-pregnant women [17]. In the present study, a decrease in time domain HRV parameters indicating shifts towards sympathetic dominance during mental challenge was markedly reduced after physical activity, including both yoga and moderate-intensity walking. However, this reduction was significantly more profound after practicing yoga compared to walking. Moreover, our results indicate a faster recovery, i.e., higher SDNN and RMSSD in the recovery phase after mental challenge in pregnant women practicing yoga regardless of the trimester. This may be explained by the fact that yoga typically encompasses more than just aerobic physical activity. Besides physical postures (asanas), yoga includes breathing (pranayama), concentration (dharana) and meditation (dhyana) exercises, which could increase parasympathetic activation and enhance stress-coping abilities. 

Several limitations of the study should be considered. This was an observational study so we could not account for all potential confounders, such as differences in maternal age and parity. It is possible that younger and nulliparous women may have had a different reactivity in response to stress. In addition, potential sociodemographic and economic differences were not considered. Furthermore, pre-pregnancy experience with yoga and other relaxation techniques were not assessed. Future research in the field, ideally in the form of well-designed and adequately powered randomized trials will be needed to evaluate the impact of previous yoga experience and its impact on stress modulation in pregnancy. Currently, no definitive conclusions on the causal relationship between yoga and the observed results can be drawn. We were also not able to assess the actual amount of physical activity in either the yoga or control groups. Participants in the yoga group attended one guided yoga session weekly throughout pregnancy, but we do not have the data on whether and/or how often they practiced at home. As a result, it is not possible to assess a possible “dose-dependent” yoga effect with more profound effects of prenatal yoga on the autonomous nervous system with a higher frequency and duration of yoga sessions. We do also not know if the participants in either group engaged in other forms of exercise during pregnancy. Future research in this field could potentially utilize digital wearable technologies to more accurately assess physical activity of the study participants. It should also be mentioned that a single yoga teacher led all the classes throughout the study, which makes generalizing the results difficult. Further studies involving different yoga teachers and different populations of pregnant women will be needed to confirm or refute our results. Nevertheless, this study adds important information on the potential modulatory effects of different forms of physical activity on stress response during pregnancy. These could influence further research in the field with potential important clinical implications.

## 5. Conclusions

Time domain HRV parameters (SDNN and RMSSD) were increased in pregnant women who practiced yoga during acute stress provoked by a standardized memory task, as well as during recovery time intervals following mental challenge. With regular practice, the effects of yoga on the autonomic nervous system response to acute stress persisted throughout all three trimesters of pregnancy. Our results suggest potentially improved stress reactivity with blunted sympathetic surges in response to acute stressful events as well as quicker recovery after psychological stress associated with regular yoga practice during pregnancy. 

## Figures and Tables

**Figure 1 sensors-24-01596-f001:**
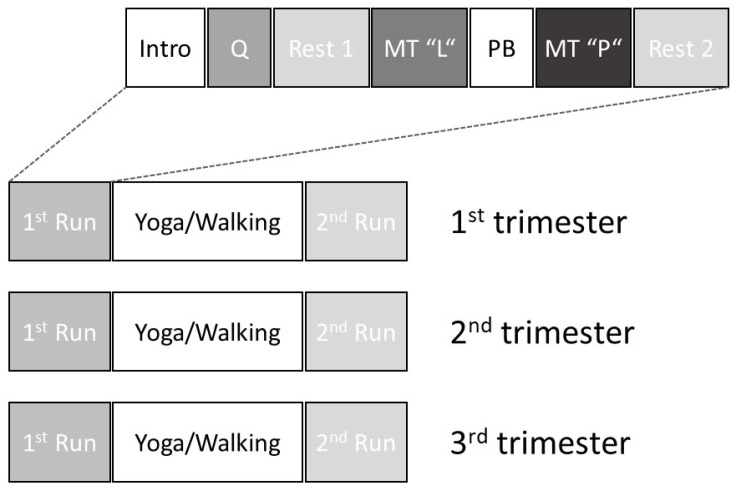
Schematic representation of the study protocol. Intro: introduction to the study and demonstration of the interventions; Q: questionnaires; Rest 1: baseline; MT “L”: memory task “learning”; PB: paced breathing at 10 min^−1^; MT “P”: memory task “practicing”; Rest 2: post-intervention recovery.

**Figure 2 sensors-24-01596-f002:**
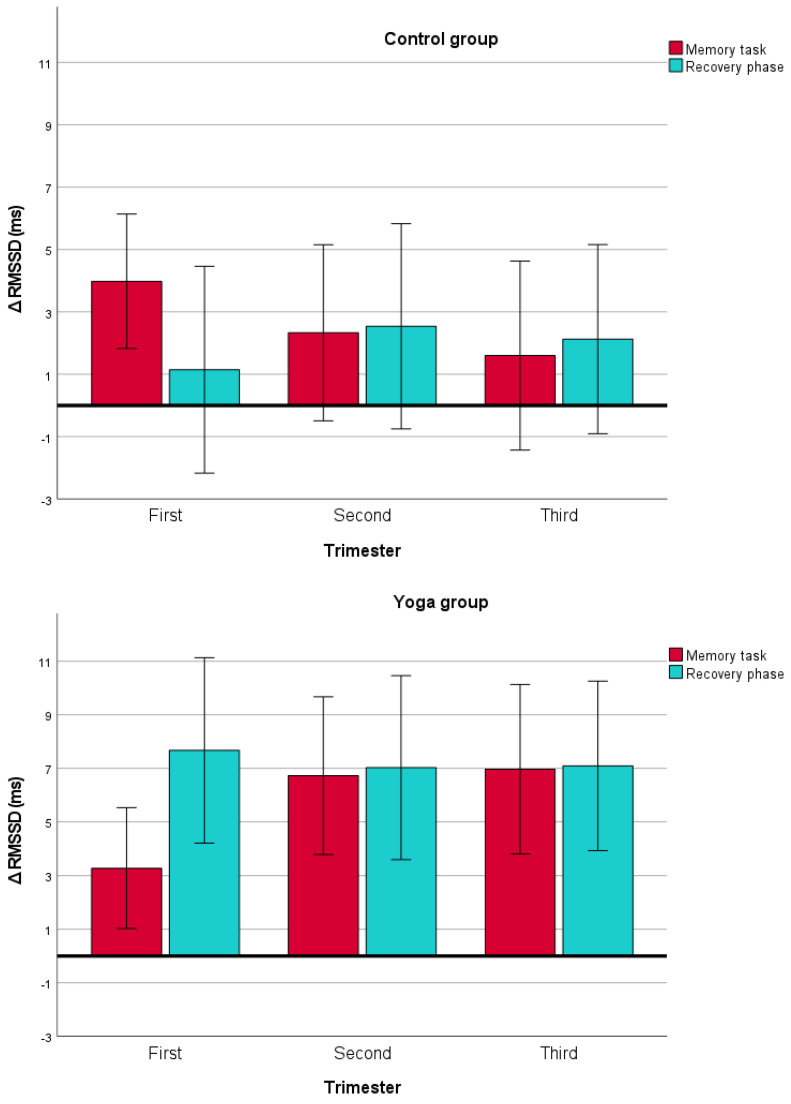
Δ RMSSD (root mean square of successive RR interval differences) with 95% confidence intervals during and immediately following a mental challenge after minus before yoga (bottom figure) vs. walking (upper figure) throughout all three trimesters of pregnancy. Note a significantly higher RMSSD values following yoga, suggesting a more pronounced decrease in the sympathetic to parasympathetic autonomic balance in the yoga group compared to controls (*F* = 9.57, *p* = 0.003).

**Table 1 sensors-24-01596-t001:** Heart rate variability parameters after minus before yoga practice (yoga group) and after minus before 30 min walk (control group) throughout pregnancy.

	Memory Task 1st Trimester	Recovery 1st Trimester	Memory Task 2nd Trimester	Recovery 2nd Trimester	Memory Task 3rd Trimester	Recovery 3rd Trimester
Δ Heart rate (bpm) *						
Yoga	−7.7 ± 5.3	−6.4 ± 4.6	−8.5 ± 6.1	−6.0 ± 5.9	−8.3 ± 7.1	−6.6 ± 6.9
Controls	−5.0 ± 5.8	−2.6 ± 5.3	−2.6 ± 4.4	−2.5 ± 4.1	−2.0 ± 4.7	−2.0 ± 4.6
Δ SDNN (ms) *						
Yoga	−0.2 ± 11.7	6.2 ± 12.5	5.0 ± 11.9	5.3 ± 12.7	1.5 ± 9.4	6.0 ± 18.6
Controls	−1.5 ± 9.1	1.4 ± 8.2	0.3 ± 7.7	1.3 ± 9.4	0.0 ± 12.1	0.2 ± 10.6
Δ RMSSD (ms) *						
Yoga	3.3 ± 6.1	7.7 ± 11.2	6.7 ± 9.2	7.0 ± 11.0	7.0 ± 9.5	7.1 ± 10.3
Controls	4.0 ± 6.8	1.1 ± 8.6	2.3 ± 7.7	2.5 ± 8.8	1.6 ± 8.7	2.1 ± 7.9
Δ ln(LF/HF) (-)						
Yoga	−0.34 ± 0.54	−0.13 ± 0.81	−0.24 ± 0.77	−0.22 ± 0.95	−0.20 ± 0.66	0.18 ± 0.76
Controls	−0.17 ± 0.70	0.02 ± 0.60	−0.02 ± 0.85	−0.15 ± 0.73	−0.08 ± 0.71	−0.06 ± 0.80

Δ values after intervention minus before intervention where intervention in the yoga group consisted of 90 min of prenatal yoga practice, intervention in the control group consisted of a 30 min walk; Δ values after intervention minus before intervention; * represents statistically significant differences (2 × 3 × 2 three-way analyses of variance; *p* < 0.05); SDNN represents the standard deviation of inter-beat intervals from which artifacts have been removed; RMSSD is the root mean square of successive R-R interval differences (inter-beat intervals between all successive heartbeats); ln(LF/HF) is the natural logarithm of the low-frequency to high-frequency power.

**Table 2 sensors-24-01596-t002:** Blood pressure variables, respiratory frequency and phase synchronization indices after minus before yoga practice (yoga group) and after minus before 30 min walk (control group) throughout pregnancy.

	Memory Task 1st Trimester	Recovery 1st Trimester	Memory Task 2nd Trimester	Recovery 2nd Trimester	Memory Task 3rd Trimester	Recovery 3rd Trimester
ΔSBP (mmHg)						
Yoga	−2.8 ± 7.9	−1.1 ± 7.0	−1.4 ± 12.0	0.5 ± 7.4	3.5 ± 11.8	0.6 ± 8.3
Controls	−5.1 ± 10.7	1.3 ± 8.4	0.8 ± 7.3	0.1 ± 7.7	−1.5 ± 9.8	0.0 ± 8.5
ΔMAP (mmHg)						
Yoga	−1.8 ± 7.3	−0.3 ± 6.2	−0.0 ± 10.3	1.3 ± 6.4	2.7 ± 9.4	1.3 ± 7.6
Controls	−3.9 ± 7.3	0.8 ± 6.6	0.8 ± 6.4	0.8 ± 6.8	−0.3 ± 8.0	−0.2 ± 7.7
ΔDBP (mmHg)						
Yoga	−1.3 ± 7.9	0.2 ± 6.9	0.3 ± 10.1	1.5 ± 5.7	1.8 ± 8.2	1.5 ± 7.2
Controls	−3.4 ± 6.3	0.2 ± 6.4	0.7 ± 6.9	1.2 ± 6.8	0.4 ± 7.7	−0.1 ± 8.3
ΔRF (min^−1^)						
Yoga	0.0 ± 2.3	0.2 ± 1.7	0.3 ± 1.5	−0.1 ± 1.5	−0.4 ± 1.9	−0.7 ± 1.5
Controls	0.0 ± 2.4	0.4 ± 1.9	0.0 ± 1.8	0.3 ± 1.5	0.5 ± 2.2	0.1 ± 1.7
Δγ_SBP × RR_ (-) *						
Yoga	0.10 ± 0.19	−0.03 ± 0.21	−0.01 ± 0.26	0.05 ± 0.23	0.03 ± 0.18	0.04 ± 0.18
Controls	−0.03 ± 0.20	−0.12 ± 0.24	0.00 ± 0.16	0.00 ± 0.20	0.03 ± 0.18	−0.01 ± 0.18
Δγ_RF × RR_ (-)						
Yoga	0.09 ± 0.28	0.01 ± 0.22	0.03 ± 0.20	0.06 ± 0.20	−0.05 ± 0.18	−0.01 ± 0.23
Controls	0.03 ± 0.18	−0.03 ± 0.18	0.05 ± 0.16	0.01 ± 0.19	0.05 ± 0.20	−0.01 ± 0.16
Δγ_RF × SBP_ (-)						
Yoga	0.11 ± 0.31	−0.03 ± 0.24	−0.06 ± 0.25	0.06 ± 0.26	−0.06 ± 0.22	−0.04 ± 0.27
Controls	−0.06 ± 0.24	−0.13 ± 0.30	0.03 ± 0.16	−0.01 ± 0.22	0.05 ± 0.22	0.00 ± 0.21

Δ values after intervention minus before intervention where intervention in the yoga group consisted of 90 min of prenatal yoga practice, intervention in the control group consisted of a 30 min walk; Δ values after intervention minus before intervention; * represents statistically significant differences (2 × 3 × 2 three-way analyses of variance; *p* < 0.05); SBP: systolic blood pressure; MAP: mean arterial pressure; DBP: diastolic blood pressure; RF: respiratory frequency; γ_SBP × RR_: synchronization index between systolic blood pressure and RR interval (inter-beat interval between all successive heartbeats); γ_RF × RR:_ synchronization index between respiratory frequency and RR interval; γ_RF × SBP:_ synchronization index between respiratory frequency and systolic blood pressure.

## Data Availability

The data presented in this study are available upon request from the corresponding author.

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
