# Peer review of "Effect of Prenatal Yoga versus Moderate-Intensity Walking on Cardiorespiratory Adaptation to Acute Psychological Stress: Insights from Non-Invasive Beat-to-Beat Monitoring"

_sensors, 2024, doi:10.3390/s24051596_

Round 1

Reviewer 1 Report

Comments and Suggestions for Authors

The paper titled "Effect of prenatal yoga versus moderate-intensity walking on cardiorespiratory adaptation to acute psychological stress: Insights from non-invasive beat-to-beat monitoring” shows a clinical study on 69 women to assess the impact of prenatal yoga on automatic nervous system stress response. 

The paper is clear and well-written, and all the references are consistence for the study. The admission criteria and the experimental test are described in detail and the results are shown with complete table and statistical test information.

However, I suggest some minor comments that can improve the paper's quality. 

- Repetition in lines 107-109, this information could be merged with the previous paragraph.

- In Table 1, the * represents statistically significant differences. Is this relative to all the specific tasks and trimesters? Maybe it is better to specify in the caption

- The conclusion paragraph should be extended to highlight the paper's contribution to that specific study.

- In the references section at line 375, there is an extra line space

Comments on the Quality of English Language

The quality of English in this document is suitable and meets the required linguistic standard.

Author Response

RESPONSE to Reviewer 1

We would like to thank for appreciating our research topic, and for all the efforts you made for improving our work!

The paper titled "Effect of prenatal yoga versus moderate-intensity walking on cardiorespiratory adaptation to acute psychological stress: Insights from non-invasive beat-to-beat monitoring” shows a clinical study on 69 women to assess the impact of prenatal yoga on automatic nervous system stress response. 

The paper is clear and well-written, and all the references are consistence for the study. The admission criteria and the experimental test are described in detail and the results are shown with complete table and statistical test information.

However, I suggest some minor comments that can improve the paper's quality. 

- Repetition in lines 107-109, this information could be merged with the previous paragraph.

The sentences on the study protocol and the measurements are now rearranged. Thank you for the useful remark.

- In Table 1, the * represents statistically significant differences. Is this relative to all the specific tasks and trimesters? Maybe it is better to specify in the caption

Correct. It relates to the 2 x 3 x 2 three-way analyses of variance used to compare differences in HRV parameters and phase synchronization indices. We have now clarified this in the table caption as suggested.

- The conclusion paragraph should be extended to highlight the paper's contribution to that specific study.

We extended the conclusions.

- In the references section at line 375, there is an extra line space

Cleared.

Reviewer 2 Report

Comments and Suggestions for Authors

This is an outstanding paper. It presents novel, potentially important and clinically relevant data for practitioners and their patients/clients.

The results are fascinating. The authors understand and employ time domain,  frequency domain, and phase synchronization analysis. 

The statistical analysis was appropriate, including three-way ANOVA, F test for continuous variables, and Chi square for categorical variables of background clinical characteristics. The effect size was estimated using the partial Eta-squared. P of 0.5 or less was considered statistically significant.

The authors acknowledge the limitations of the study. It is an observational study, not a randomized controlled trial, and all potential cofounders could not be accounted for. Therefore, caution must be exercised if causality is inferred.

My only suggestion would be for the authors to consider suggestions for further studies that could address these limitations.

Overall, a fascinating and clinically useful study.

Author Response

RESPONSE to Reviewer 2

We would like to thank for appreciating our research topic, and for all the efforts you made for improving our work!

This is an outstanding paper. It presents novel, potentially important and clinically relevant data for practitioners and their patients/clients.

The results are fascinating. The authors understand and employ time domain,  frequency domain, and phase synchronization analysis. 

The statistical analysis was appropriate, including three-way ANOVA, F test for continuous variables, and Chi square for categorical variables of background clinical characteristics. The effect size was estimated using the partial Eta-squared. P of 0.5 or less was considered statistically significant.

The authors acknowledge the limitations of the study. It is an observational study, not a randomized controlled trial, and all potential cofounders could not be accounted for. Therefore, caution must be exercised if causality is inferred.

My only suggestion would be for the authors to consider suggestions for further studies that could address these limitations.

We thank the reviewer for his insight. We have added some suggestions for future research in the field to the discussion.

Overall, a fascinating and clinically useful study.

Thank you so much!

Reviewer 3 Report

Comments and Suggestions for Authors

This manuscript makes use of recent trial data to analyze potential correlations between prenatal yoga on HRV in pregnant women, as compared to walking.

The study design makes sense but some critical components need clarification. For example, the Methods do not make clear whether this is a study of women already doing yoga, or whether yoga is an intervention and they were confirmed to not be doing yoga prior. Same with the counterbalanced walking intervention. It is implied, but it should be stated explicitly to ensure effects are not self-selection artifacts.

It would be good to address the age difference between the groups. Older women (late 20s v early 20s) might well have more experience and stress-coping skills, or be more financially stable, etc., so that the effects between groups could be accounted for by other things than yoga. If there is overlap in ages of individuals in each group, then a regression of effect or HRV vs age could help to disambiguate the effects.

Finally, there is a good deal of discussion (second sentence of Discussion as a key example) about this demonstrating effects on stress and resilience. HRV is not stress and resilience. The correlation is real but much weaker and more complex than is implied here. It is not reasonable to assume subjective states like resilience from small changes in HRV. This needs to be clarified.

Author Response

RESPONSE to Reviewer 3

We would like to thank for appreciating our research topic, and for all the efforts you made for improving our work!

This manuscript makes use of recent trial data to analyze potential correlations between prenatal yoga on HRV in pregnant women, as compared to walking.

The study design makes sense but some critical components need clarification. For example, the Methods do not make clear whether this is a study of women already doing yoga, or whether yoga is an intervention and they were confirmed to not be doing yoga prior. Same with the counterbalanced walking intervention. It is implied, but it should be stated explicitly to ensure effects are not self-selection artifacts.

We agree this is a very important point. Previous yoga experience was not considered an exclusion criterion. This has now been clarified in the Methods section. We have also discussed this among the study’s limitations.

It would be good to address the age difference between the groups. Older women (late 20s v early 20s) might well have more experience and stress-coping skills, or be more financially stable, etc., so that the effects between groups could be accounted for by other things than yoga. If there is overlap in ages of individuals in each group, then a regression of effect or HRV vs age could help to disambiguate the effects.

We agree. Given the observational nature of the study, several factors other than yoga could have influenced our results. This is why no claims on causality can be made. This has been clearly stated. However, given the relatively small size of the two study groups, we decided not to perform subgroup analyses as this would inevitably lead to higher risk of type I statistical error. As mentioned in the discussion, future research, ideally in form of randomized trials, will be needed to assess the potentially causal nature of associations between yoga practice and improved stress reactivity.

Finally, there is a good deal of discussion (second sentence of Discussion as a key example) about this demonstrating effects on stress and resilience. HRV is not stress and resilience. The correlation is real but much weaker and more complex than is implied here. It is not reasonable to assume subjective states like resilience from small changes in HRV. This needs to be clarified.

We agree. We have rewritten parts of discussion and conclusions in order not to over-interpret HRV results.

Round 2

Reviewer 3 Report

Comments and Suggestions for Authors

This revised MS is improved in clarity and removes several ambiguities from the previous version. My only suggestion is that the Y axes of 2's two plots be the same to facilitate comparison. This is a minor proofing point.

Author Response

Many thanks for this note! We have adjusted the graphics.